# Societal benefits of halving agricultural ammonia emissions in China far exceed the abatement costs

Xiuming Zhang [1], Baojing Gu [2✉], Hans van Grinsven [3], Shu Kee Lam [1], Xia Liang[1], Mei Bai[1] & Deli Chen [1✉]

Mitigating agricultural ammonia ($NH_3$) emissions in China is urgently needed to avoid further damage to human and ecosystem health. Effective and feasible mitigation strategies hinge on integrated knowledge of the mitigation potential of $NH_3$ emissions and the associated economic costs and societal benefits. Here we present a comprehensive analysis of marginal abatement costs and societal benefits for $NH_3$ mitigation in China. The technical mitigation potential of agricultural $NH_3$ emissions is 38–67% (4.0–7.1 Tg N) with implementation costs estimated at US$ 6–11 billion. These costs are much lower than estimates of the overall societal benefits at US$ 18–42 billion. Avoiding unnecessary fertilizer use and protein-rich animal feed could provide 30% of this mitigation potential without additional abatement costs or decreases in agricultural productivity. Optimizing human diets with less animal-derived products offers further potential for $NH_3$ reduction of 12% by 2050.

[1] School of Agriculture and Food, The University of Melbourne, Melbourne, VIC 3010, Australia. [2] College of Environmental and Resource Sciences, Zhejiang University, Hangzhou 310058, PR China. [3] PBL Netherlands Environmental Assessment Agency, PO BOX 30314, 2500 GH The Hague, The Netherlands. ✉email: bjgu@zju.edu.cn; delichen@unimelb.edu.au

Anthropogenic ammonia ($NH_3$) emissions, primarily from agriculture, have adversely affected environmental quality, including air pollution, soil acidification, eutrophication of water bodies, and led to tremendous damage to human health and ecosystem health[1,2]. The cost of damage associated with agricultural $NH_3$ emissions was estimated at US dollars (US\$) 55–114 billion in the European Union (EU) in 2008, with the largest contribution due to increased human mortality from exposure to $NH_3$-containing aerosols[3,4]. In the United States (US), annual health costs due to $NH_3$ emissions were estimated at US\$69–180 billion in 2011[5].

Mitigating $NH_3$ emissions has attracted much attention in high-income countries. For example, the Gothenburg Protocol was signed in 1999 to control long-range transboundary transport of air pollutants among member countries within the United Nations Economic Commission for Europe. Following the Gothenburg protocol, the EU adopted the first National $NH_3$ Emission Ceilings directive (2001/81/EC) in 2001[6]. The efficacy and costs of $NH_3$ abatement and their climate co-benefits were evaluated in 2015[7], focusing on European countries. To date, only a few countries have estimated their national $NH_3$ mitigation potential and associated costs and benefits (Table 1).

China is the world's largest emitter of $NH_3$ (9–13 Tg N year$^{-1}$ in the 2010s), with over 80% contributed by agriculture[8,9]. Low fertilizer nitrogen (N) use efficiency (NUE) and poor animal waste management have resulted in enormous $NH_3$ emissions in China[8,10]. Worse still, regional $NH_3$-related pollution is enhanced due to the increasing decoupling between crop and livestock production systems[11]. In recent years, frequent smog events with high concentrations of $PM_{2.5}$ (fine particulate matter $< 2.5\,\mu m$) in China have triggered both public anxiety and concerns of the Chinese government[12]. A substantial proportion of $PM_{2.5}$ pollution was caused by aerosol formation driven by $NH_3$ emissions[13,14]. Studies have suggested that the current clean air policy for reductions in primary $PM_{2.5}$, sulfur dioxide ($SO_2$), and nitrogen oxides ($NO_x$) has limitations, and that $PM_{2.5}$ pollution can be cost-effectively controlled only if $NH_3$ emissions are abated as well as those of $SO_2$ and $NO_x$[15–17]. Studies have also suggested that many $NH_3$ abatement techniques may simultaneously reduce agricultural methane ($CH_4$) and nitrous oxide ($N_2O$) emissions, bringing co-benefits for agricultural greenhouse gas (GHG) mitigation[7,18–20]. However, $NH_3$ emission reduction in China may worsen the adverse impact of acid rain on crops and forests by increasing rainfall acidity[21,22].

To date, China has not yet formulated or implemented policies to reduce $NH_3$ emissions[23], although there are many available measures to reduce $NH_3$ emissions from agriculture, most of which have been validated and adopted in the EU and North America[7,24]. Many $NH_3$ abatement measures have not been widely practiced in China and their implementation costs and the impacts on agricultural GHG emissions have not been assessed. Given that poor smallholder farmers still dominate China's agricultural production and that agricultural N pollution is severe[25], it is crucial to identify feasible and cost-effective $NH_3$ abatement measures for Chinese agriculture.

A national systematic assessment of $NH_3$ mitigation potential, and the associated costs and societal benefits, is urgently needed for China to establish cost-effective mitigation strategies and targets. To fill the knowledge gap, this study builds an integrated $NH_3$ mitigation assessment framework (Supplementary Fig. 1) with the combination of Coupled Human And Natural Systems (CHANS), GAINS, Weather Research and Forecasting-Community Multiscale Air Quality (WRF-CMAQ), and exposure–response models to: (1) identify feasible $NH_3$ abatement options and to estimate the agricultural $NH_3$ mitigation potential and the associated implementation costs and societal benefits; (2) determine mitigation priorities and strategies for China; and (3) to explore optimal $NH_3$ mitigation pathways for the next 30 years (2020–2050) using scenario analysis and cost-benefit assessment. We find that the relative $NH_3$ mitigation potential in China is twice that in Europe. The overall societal benefits of agricultural $NH_3$ mitigations in China far exceed the abatement cost and increase when including the synergy with reduction of GHG emissions.

## Results and discussion

**$NH_3$ mitigation potential, abatement costs, and societal benefits.** For cropping systems $NH_3$ abatement measures include reductions of urea-based fertilizer, promotion of enhanced efficiency N fertilizer (EENF), and deep placement of fertilizer (Supplementary Table 1). The $NH_3$ mitigation potential of crop production is around 2.0–3.4 Tg N year$^{-1}$ at an abatement cost of US\$1.9–3.4 billion. The three major staple crops in China have the largest $NH_3$ mitigation potential at 460–954 Gg N for maize, 516–684 Gg N for rice, and 446–731 Gg N for wheat. The large reduction potential is mainly due to large sowing areas and poor fertilization practices. The production of vegetables and fruits consumes one-third of total synthetic N fertilizer use in China, and their $NH_3$ mitigation potentials are estimated at 30–55% (269–493 Gg N) and 20–40% (118–235 Gg N), respectively. Unit abatement costs (US\$ ha$^{-1}$ year$^{-1}$, Table 2) for cash crops (sugar, fruits and, vegetables) are higher than those for staple crops because the production of cash crops is more intensive, requiring higher inputs of manpower, fertilizer, and financial resources[26].

For livestock production systems $NH_3$ abatement measures include manipulation of feed rations, improved housing facilities

**Table 1 $NH_3$ mitigation potential and costs in different countries.**

| | NH₃ emission (Gg N year$^{-1}$) (2000s) | Health damage cost (US\$ billion year$^{-1}$) (2000s) | Mitigation potential (%) (2020) | Unit abatement cost (US\$ kg$^{-1}$) (2020) | Total abatement cost (US\$ million) |
|---|---|---|---|---|---|
| Denmark[30] | 43 | 0.6[a] | 7–12 | 1.1–4.0 | 1.9–7.1 |
| Netherlands[30] | 109 | 4.1[a] | 7–11 | 0.3–3.5 | 1.4–23.9 |
| Germany[30] | 467 | 15.2[a] | 25–39 | 1.6–2.6 | 83–377 |
| EU27[3,4,30,67] | 3421 | 55–114[b] | 20–35 | 1.2–3.5 | 821–4129 |
| USA[37] | 3046 | 69–180[c] | NA[d] | 8.0 | NA |
| Canada[24] | 421 | NA | 29 | NA | NA |
| China (this study) | 12,277 | 44–115[e] | 38–67 | 0.8–2.1 | 6146–11,198 |

[a]Derived from Brink and Van Grinsven[68].
[b]Derived from Van Grinsven et al.[3].
[c]Derived from Goodkind et al.[5].
[d]NA means data "Not Available" or "Not Applicable."
[e]Calculated based on the methods in Gu et al.[59] and updated VSL from Giannadaki et al.[1].

**Table 2 Ammonia mitigation potential and costs for agricultural products (2020).**

| | Mitigation Potential (%) | Absolute Reduction (Gg NH$_3$-N year$^{-1}$) | Unit cost (US\$ ha$^{-1}$ year$^{-1}$) or (US\$ LU$^{-1}$ year$^{-1}$ [a]) | Cost-effectiveness (US\$ per kg NH$_3$-N) | Total cost (US\$ billion) |
|---|---|---|---|---|---|
| For cropland | 39–68 | 1977–3420 | 12–21 | 0.6–1.7 | 1.9–3.4 |
| Rice | 55–73 | 516–684 | 14–18 | 0.6–1.2 | 0.4–0.6 |
| Wheat | 51–83 | 446–731 | 16–27 | 0.5–1.3 | 0.4–0.6 |
| Maize | 39–81 | 460–954 | 11–22 | 0.4–2.0 | 0.4–0.9 |
| Beans | 22–38 | 9–15 | 1–2 | 0.7–1.2 | 0.01–0.01 |
| Tubers | 25–46 | 55–101 | 4–8 | 0.2–1.8 | 0.0–0.1 |
| Cotton | 35–83 | 45–106 | 11–25 | 0.7–2.2 | 0.0–0.1 |
| Oil crops | 27–49 | 25–44 | 2–4 | 0.7–2.0 | 0.03–0.05 |
| Sugar crops | 45–70 | 36–57 | 28–44 | 0.5–2.7 | 0.0–0.1 |
| Fruits | 20–40 | 118–235 | 18–35 | 0.9–3.4 | 0.2–0.4 |
| Vegetables | 30–55 | 269–493 | 15–27 | 0.6–2.2 | 0.3–0.6 |
| For livestock | 37–65 | 2051–3635 | 11–20 | 1.2–2.7 | 4.2–7.8 |
| Dairy cattle | 36–61 | 149–251 | 30–55 | 2.1–6.6 | 0.5–1.0 |
| Beef cattle | 37–61 | 300–499 | 44–86 | 2.4–7.7 | 1.2–2.3 |
| Sheep and goat | 32–62 | 277–533 | 6–15 | 0.3–1.3 | 0.2–0.3 |
| Sow | 40–68 | 205–347 | 23–37 | 1.4–3.7 | 0.5–0.8 |
| Hog | 41–69 | 633–1061 | 8–14 | 1.1–3.3 | 1.2–2.1 |
| Laying hen | 37–73 | 232–453 | 10–19 | 0.5–2.0 | 0.2–0.5 |
| Other poultry | 35–72 | 189–390 | 3–6 | 0.9–3.7 | 0.3–0.7 |
| Rabbit | 42–57 | 36–48 | 7–11 | 0.4–1.0 | 0.01–0.03 |
| Horse/donkey/ mule | 24–41 | 31–53 | 4–9 | 0.9–3.0 | 0.05–0.09 |
| Camel | 6–16 | 0.0–0.1 | 0–0 | 0.1–0.5 | 0.0–0.0 |
| Total | 38–67 | 4028–7055 | NA[b] | 0.8–2.1 | 6.1–11.2 |

[a]LU conversion coefficients used in this study are derived from Bai et al.[27], namely, 1 head of dairy cattle, beef cattle, pig, sheep and goat, layer, and broiler equal 1.0, 0.50, 0.35, 0.10, 0.012 and 0.007 LU, respectively
[b]NA means data "Not Available" or "Not Applicable."

and manure management practices (Supplementary Table 2). The NH$_3$ mitigation potential of livestock production is around 2.1–3.6 Tg N year$^{-1}$ at an abatement cost of US\$4.2–7.8 billion. The pig industry in China has the largest NH$_3$ mitigation potential at 838–1408 Gg N, followed by poultry farming (421–843 Gg N) and cattle production (448–751 Gg N). Livestock units (LUs), a metric used in this study to compare the unit abatement costs between different animal types (Table 2) on the basis of the feed requirement of each type of animal[27]. Generally, the unit abatement costs differ notably among animal types. For instance, the unit abatement cost is the highest for beef cattle (US\$44–86 LU$^{-1}$ year$^{-1}$) while it is only US\$8–14 LU$^{-1}$ year$^{-1}$ for hogs. Differences are due to inherent disparities in animal feed, digestibility and farming practices[28]. The total abatement cost for the cattle industry is the highest (US\$1.7–3.3 billion), followed by pig (US\$1.6–2.9 billion) and poultry (US\$0.6–1.2 billion) farming. For other types of livestock abatement costs are relatively low owing to their smaller NH$_3$ emission rates and smaller animal populations. When mitigation options are combined for different crops and animal types, total agricultural NH$_3$ mitigation potential is estimated at 38–67% (4.0–7.1 Tg N) of total NH$_3$ emissions, with implementation costs estimated at US\$6–11 billion, equivalent to 0.04–0.08% of the national GDP of China.

The societal benefits of NH$_3$ emission reduction were also quantified for comparison with the implementation costs of abatement measures. The mitigation of NH$_3$ emissions by 38–67% could reduce PM$_{2.5}$ concentrations by 8–20%, and avoid premature mortalities by 90–240 thousand people with health benefits at US\$10–26 billion. Ecosystem benefits due to NH$_3$ mitigation in terms of avoided soil acidification and water eutrophication are estimated at US\$10–17 billion. Further, NH$_3$ mitigation could simultaneously reduce agricultural GHG emissions by 9–35% (101–385 Tg CO$_2$ equivalents (CO$_2$-eq)), and

generate climate benefits of US\$1–3 billion. However, reductions of NH$_3$ emissions might increase the acidity of precipitation and cause an economic loss of US\$4–7 billion. Although acid rain damage partly offsets the benefit of NH$_3$ mitigation, the overall societal benefits of NH$_3$ control (US\$18–42 billion) still far outweigh the abatement costs (US\$6–11 billion), suggesting that mitigation of agricultural NH$_3$ emissions could generate net economic benefits (NEBs) on a national scale.

**NH$_3$ mitigation priority and strategies for China.** The marginal abatement cost curve (MACC) can be used to support decision making in prioritizing strategies for NH$_3$ mitigation[15,29]. Using data presented in Table 2, a bottom-up MACC was constructed (Fig. 1), which plots the cumulative NH$_3$ emission-reducing potential of measures with increasing implementation cost per unit of NH$_3$ emission reduction. The MACC illustrates that the agricultural sector offers an average reduction potential of 5.5 Tg NH$_3$–N at a total cost of US\$8.3 billion. A reduction of 1.6 Tg NH$_3$–N (30% of the total reduction) is potentially available at a negative cost (cost-saving) for the agricultural sector by preventing unnecessary use of N fertilizer and protein-rich animal feed. These savings can be used to cover the implementation cost of the next incremental reduction of 1.7 Tg NH$_3$–N. As a result, 3.3 Tg NH$_3$–N (60% of the total reduction) can be abated at zero cost. Overall, a 90% reduction could be achieved at an average cost below US\$1.2 per kg NH$_3$–N, which is the estimated threshold of abatement cost in the EU$_{27}$ to meet the targets of the Thematic Strategy on Air Pollution[30].

The MACC highlights the importance of reducing synthetic N fertilizer use, coupled with improved animal feeding practices, as potential foci for mitigating China's agricultural NH$_3$ emissions. We found that mitigating China's agricultural NH$_3$ emissions

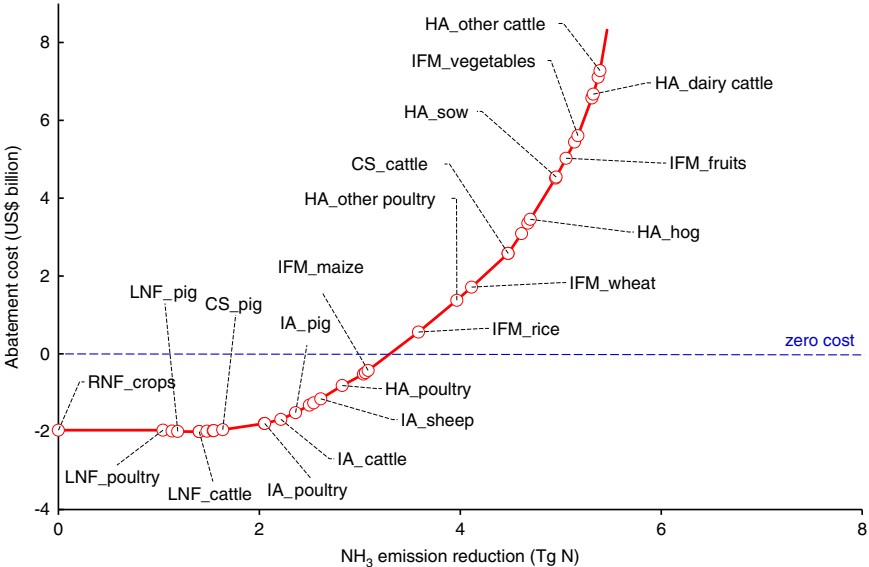

**Fig. 1 Marginal abatement cost curve of agricultural NH₃ emissions in China.** The red dots represent the introduction of specific mitigation options. Only measures with a significant reduction potential (>0.1 Tg NH₃–N) are labeled; the blue dotted line represents abatements cost = 0; the RNF is the direct reduction of synthetic N fertilizer (mainly urea) use on croplands without yield loss, since currently the high application rates of N fertilizer in China far exceed crop demand[31]; LNF represents low N feeding options without decreasing animal productivity; CS represents covered storage measures; IA represents improved manure application options; HA represents housing adaptation options; IFM means improved fertilization management including right source, right place, right time of fertilization and irrigation. Note that in this figure we only give the weighted mitigation potential and weighted cost of the proposed mitigation options for different crops or animals. Details about combinations of mitigation measures are summarized in Supplementary Table 7. Source data are provided as a Source Data file.

should start with the reduction of urea-based N fertilizer (RNF) in cropland by 20%. This option could offer 27% of mitigation potential for cropland and save fertilizer costs of US$2.0 billion without yield loss. In general, fertilizer N application in China has far exceeded the crop demand[31]. Studies have proven that reducing current N applications by 30–60% with optimum N management would still maintain crop yields while saving unnecessary economic expenditure for farmers[19,31]. Also, the reduction of protein (N) rich feed (LNF) could reduce N losses from excreta and result in a decrease in 0.5 Tg of NH₃–N emission without undermining animal productivity. The next priority mitigation measures are covered storage (CS) of pig and poultry manures, followed by improved application (IA) of animal manure to cropland. These measures are able to reduce losses of 1.2 Tg of NH₃–N with a cost of US$1.1 billion for manure processing. In addition, housing adaptation (HA) for poultry can reduce another 0.3 Tg of NH₃–N emission at a cost of US$0.6 billion. Improved N fertilizer management (IFM) for three major staple crops can reduce NH₃–N emission by 1.2 Tg of at a cost of US$2.6 billion. In practice, the net cost of IFM may be smaller because the increased crop quality and yield may partly compensate for the investment costs (e.g., equipment such as subsurface injectors) and operational costs (e.g., material, labor, and energy). Advanced housing systems for pigs and cattle (e.g., air-scrubbing techniques) can be very effective for reducing NH₃ emissions (up to 80%) but are also costly (US$10–30 per kg NH₃–N mitigated).

**NH₃ mitigation pathways in 2020–2050.** Scenario analysis and cost-benefit assessment guide the optimization of NH₃ mitigation strategies and pathways in the future. In this study, one baseline scenario of business as usual (BAU) and four mitigation scenarios (DIET, NUE, REC, and ALL) toward 2050 are analyzed, which comprise a range of packages of mitigation options (see Table 3 for details) to explore optimal mitigation pathways. The simulated NH₃ emission trends (Fig. 2) for the next 30 years

(2020–2050) reveal that there would be substantial NH₃ mitigation potential with broad welfare benefits (Fig. 3).

Under the BAU scenario total agricultural NH₃ emissions in China are estimated to first increase from 10.9 Tg N in 2015 to 12.1 Tg N in 2035 because of a growing and changing food demand for China's increasing and wealthier population[27]. The emission would then slightly decrease to 11.9 Tg N following a decrease in China's population toward 2050[32]. NH₃ emission from synthetic N fertilization is expected to remain stable during this period considering the national "Zero-growth Action Plan" for chemical fertilizer use[33]. The major cause of increased NH₃ emission is the rising livestock production to meet the growing demand for animal products both in total and per capita terms[27,34].

In contrast, the DIET scenario assumes optimizing human dietary structure (transitioning toward more plant-based diets) to reduce the animal-based food N to 40%, which is in line with Chinese dietary guidelines[35]. The increased human consumption of plant-based food N will shorten the food chain and decrease food-feed competition from decreased livestock farming. Decreased livestock production (meat, eggs, and milk) in DIET reduces the demand for crop production by 20–30% relative to the BAU scenario, which could reduce NH₃ emission by 21% by 2050 (Fig. 2).

Based on the proposed improvement in NUE in crop and livestock production systems by adopting advanced farming practices, or techniques as identified in this study, agricultural NH₃ emissions are projected to decline from 11.9 to 8.8 Tg N in 2050 under the NUE scenario (Fig. 3). This scenario could decrease synthetic N fertilizer use by 13 Tg N, decreasing NH₃ emission from cropping systems by 39%. In addition, NH₃ emission from livestock systems could be reduced by 1.9 Tg N through improved animal feed and waste management.

Cropland in China is poorly coupled with its livestock production systems[11]. The REC scenario aims to reconnect the two agricultural subsystems by increasing the recycling of manure

**Table 3 Proposed NH₃ mitigation pathways with key indicators.**

| Scenario | Description | Key indicators in 2050 | Related options | Main consequence or effect |
|---|---|---|---|---|
| BAU | Only consider current policies and national plans without any further intervention. Consumption of meat and other animal products is growing rapidly | $Ratio_a = 60\%$; $NUE_c = 30\%$; $NUE_a = 15\%$; $REC_a = 30\%$; $REC_s = 28\%$; $REC_h = 23\%$ | None | Substantially increased crop production for animal feed and meat production to feed growing and wealthier population |
| DIET | Optimize human dietary structure by cutting consumption of animal products following the Chinese Dietary Guidelines. | $Ratio_a = 40\%$; $NUE_c = 30\%$; $NUE_a = 15\%$; $REC_a = 30\%$; $REC_s = 28\%$; $REC_h = 23\%$ | Human dietary change | Increased land area released from the reduction of growing animal feed; reduced net land requirement for crop and livestock production under DIET scenario |
| NUE | Boost N use efficiency through balanced N fertilizer application: cropping system with 4 R fertilization management; Livestock production system with feeding and manure management | $Ratio_a = 60\%$; $NUE_c = 40\%$; $NUE_a = 25\%$; $REC_a = 30\%$; $REC_s = 28\%$; $REC_h = 23\%$ | C1–C7 L1–L13 L18 | Reduced fertilizer consumption; reduced manure N loss from livestock production |
| REC | Cut agricultural waste by improving recycling of livestock manure, crop residue and human waste in agroecosystems to partially substitute synthetic fertilizer nitrogen (N) input and increase crop yield | $Ratio_a = 60\%$; $NUE_c = 30\%$; $NUE_a = 15\%$; $REC_a = 60\%$; $REC_s = 40\%$; $REC_h = 50\%$ | L14–L17 | Reduced use of chemical fertilizer N; more manure N being recycled to the field |
| ALL | Combination of Diet, NUE and REC, namely, LCP diet, improvement in fertilization and manure management, recycling manure N to cropland | $Ratio_a = 40\%$; $NUE_c = 40\%$; $NUE_a = 25\%$; $REC_a = 60\%$; $REC_s = 40\%$; $REC_h = 50\%$ | Human dietary change C1–C9 L1–L18 | Reduced livestock production; reduced use of chemical fertilizer; reduced manure N loss; more manure N being recycled to the field |

$Ratio_a$ is the share of animal products in human diet; Nitrogen Use efficiency (NUE) is defined as the N output in useful products as percentage of the total N input; $NUE_c$, $NUE_a$ represent the N use efficiency in cropland and animal production, respectively; $REC_a$, $REC_s$, $REC_h$ refer to the recycling ratio of animal excretion, crop straw and human waste, respectively.

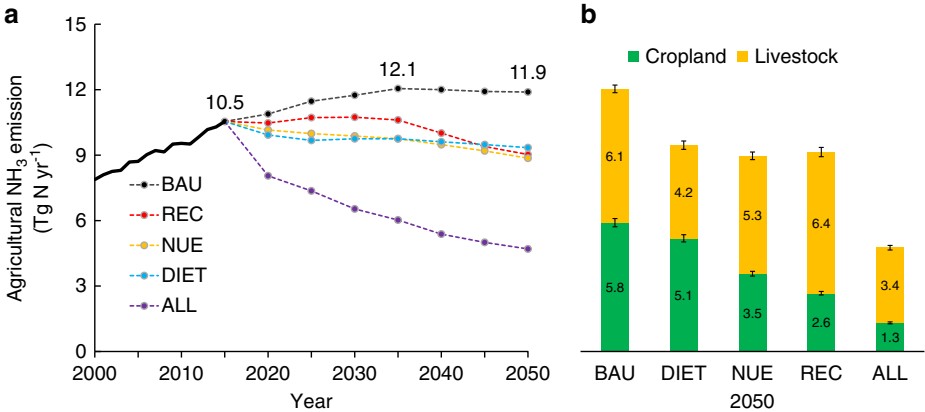

**Fig. 2 Agricultural NH₃ emissions under different scenarios in 2050.** Agricultural NH₃ emissions by different scenarios (**a**) and source contribution (**b**) in 2050. BAU refer to business as usual scenario; REC represents the scenario of manure recycling to cropland; NUE is the scenario of improving N use efficiency both in cropland and livestock production systems; DIET is the scenario of optimizing human dietary structure; ALL scenario is the combination of all NH₃ mitigation measures in REC, NUE and DIET scenario, the error bars denote the 95% confidence intervals of the specific emissions. Source data are provided as a Source Data file.

to croplands. The total excretion N generated by livestock production was 13.4 Tg N in 2015 and is estimated to reach 18.2 Tg N in 2050 under the REC scenario. Simultaneously, the total N demand of crops in China is estimated to be 25.0 Tg N in 2050, suggesting that animal excretion N would be within the carrying capacity of cropland in China. Under the assumption that 60% of

manure N is recycled to croplands, the REC scenario could save 10.9 Tg chemical N fertilizer and reduce NH₃ emission by 24% (3.3 Tg N) in 2050. The abatement cost is estimated at US$3.8 billion, acknowledging the considerable socio-economic barriers of relocation and adaptations of the livestock supply chain and transport infrastructure[3].

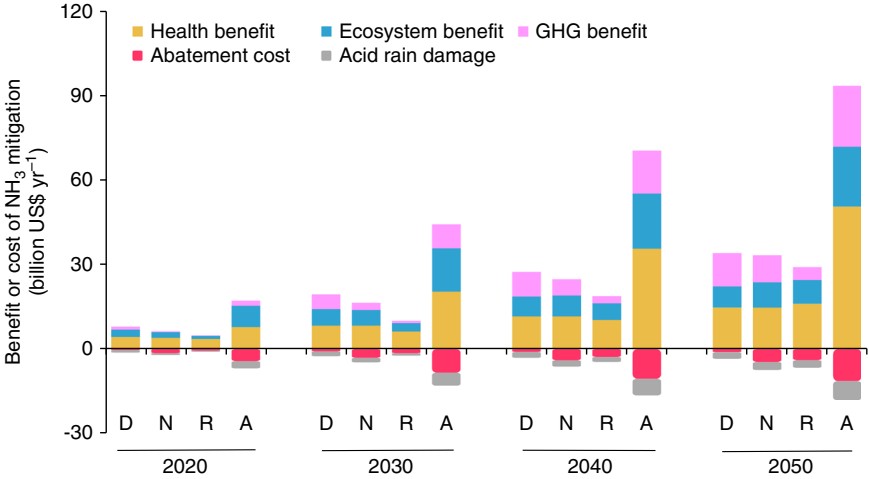

**Fig. 3 Costs and benefits of NH3 mitigation during 2020–2050 in China.** D, N, R, A refer to the DIET, NUE, REC and ALL scenarios, respectively; each scenario features different mitigation pathways and generates different mitigation costs and benefits. Source data are provided as a Source Data file.

To achieve the most ambitious mitigation target, the ALL scenario combines all the mitigation options identified in DIET, NUE, and REC scenarios. The estimated mitigation potential of the ALL scenario is 7.2 Tg $NH_3$–N (61% reduction relative to BAU), of which an 80% reduction is achieved by improved agricultural management and technical solutions, while the remaining 20% reduction is due to decreasing the consumption of animal products. The ALL scenario suggests that after achieving the technical mitigation potential, reducing consumption of animal products could offer a further 12% of $NH_3$ mitigation potential by 2050.

However, the abatement costs vary among the four mitigation scenarios: The DIET scenario has the smallest mitigation cost (<US$1.0 billion) given that shifting to sustainable diets mainly depends on cost-free adjustment of consumers' preference and behaviors. The ALL scenario has the greatest abatement cost (up to US$11 billion in 2050) because it requires comprehensive and coordinated actions.

The benefits from improved human and ecosystem health increase with the extent of $NH_3$ mitigation for all mitigation scenarios. The NUE, DIET, and REC scenarios have similar mitigation potentials and human and ecosystem health benefits of US$22–26 billion while the ALL scenario generates the highest health benefit of US$72 billion in 2050 (Fig. 3). The agricultural GHG emissions vary between the $NH_3$ mitigation scenarios. The DIET, NUE, and ALL scenarios can significantly reduce agricultural GHG emissions and generate positive climate benefits, while REC slightly decreases GHG emissions because manure addition to soil increases GHG ($N_2O$) emission over time[36], but that could be partially offset by indirect $N_2O$ mitigation. The reduction of agricultural $NH_3$ emissions may aggravate acid rain over China and may result in economic loss under all mitigation scenarios (Fig. 3). Nevertheless, the overall societal benefits under all four mitigation scenarios still exceed the corresponding abatement costs and result in net national social welfare.

**International comparison.** The $EU_{27}$ (27 countries of the EU) has a history of $NH_3$ mitigation over about two decades[6], while Canada has just revealed its national mitigation potential[24]. In the USA, several studies have indicated that agricultural $NH_3$ emissions are an important driver of $PM_{2.5}$ pollution causing huge health costs[5,37].

The current average $NH_3$ mitigation potential of China (53%) is around twice that of the $EU_{27}$ (24%)[30] and Canada (29%)[24]

(Table 1). This is not surprising because China has the highest quantities of $NH_3$ emissions in the world but has not yet implemented mitigation policies. There are two main reasons for the high $NH_3$ emissions currently in China. First, the total amount of fertilizer applied to Chinese croplands has increased more than threefold since 1980, accounting for one-third of global synthetic N fertilizer consumption[10]. The excessive N input, poor farming practices and small farm size have led to a low NUE and high $NH_3$ losses in Chinese cropping systems[25]. Second, the livestock population increased from 142 to 441 million LUs, almost tripling between 1980 and 2010[27]. The rapid growth in intensive industrial livestock production and unsustainable management practices arising from the unbalanced spatial distribution of livestock farms to poor animal house cleansing and manure handling have resulted in large waste of nutrients in manure[11,27], which further increases $NH_3$ emissions from animal husbandry.

The weighted unit abatement cost for China is estimated to be US$0.8–2.1 per kg $NH_3$–N, which is lower than that of the $EU_{27}$ at US$1.2–3.5 per kg $NH_3$–N[38]. This discrepancy is attributed to the large differences in farm sizes, labor costs, and agricultural mechanization levels. Recent studies indicate that larger-scale farms are more efficient in fertilizer use, labor, and professional management than smaller ones[25]. However, smallholder farming dominates the agricultural landscape in China[39]. Land fragmentation and small farm size in China reduce the efficiency of machinery and services[25,40]. However, the unit labor cost in China is around one-sixth of the EU, which can partly explain the lower implementation cost for the early stage of $NH_3$ mitigation (defined as the implementation for the first 60% of mitigation potential) in China as compared to the $EU_{27}$.

The low-hanging fruit for achieving reductions in $NH_3$ reductions is the direct reduction of urea-based fertilizer use and protein-rich feeding in animal production. The remaining mitigation will become increasingly more expensive due to the greater requirements of technologies and equipment, with a marginal abatement cost in the range of US$2–15 per kg $NH_3$–N mitigated in China (Fig. 1). The transition to large-scale and mechanized agriculture in China is restricted by inherent social barriers and weak technical foundation[26], which takes time and effort to overcome.

**Policy implications.** To clean up the air, Chinese governments have already made major efforts to reduce anthropogenic $SO_2$

and $NO_x$ emissions, which have declined by 41% and 34%, respectively, from their peaks to 2019[41]. Although continuing the stringent policies to reduce $SO_2$ and $NO_x$ emissions could further improve air quality, and may partially offset the effects of $NH_3$ mitigation, studies have suggested that current policies are not sufficient or cost-effective in achieving the targets of clean air in China[16,23,42]. A recent study has found that reducing livestock $NH_3$ emissions would be highly effective in reducing $PM_{2.5}$ during severe winter haze events[43]. Our quantitative assessments of the implementation cost and societal effects of $NH_3$ mitigation in China further demonstrate that $NH_3$ mitigation could generate net societal benefits, even though it may worsen regional acid rain. Therefore, coordinated mitigation of multiple air pollutants ($SO_2$, $NO_x$, and $NH_3$) should be implemented to more rationally and effectively achieve the dual benefits of protecting human and ecosystem health in China at both national and regional scales[21].

For farm holders, strategic designs of cost-effective mitigation pathways are needed. The aforementioned cheap and easy mitigation options (direct reduction of N fertilizer use and improved animal feeding practices) should be introduced first to pick the low-hanging fruit of $NH_3$ mitigation in China. The remaining mitigation measures (e.g., HA and manure handling systems) are expensive due to the higher requirements of the investments in technologies and infrastructures. It is necessary to increase government support (e.g., technical guidance and training) and subsidies (e.g., enhanced efficiency fertilizers and agricultural machinery) to encourage farmers to adopt these mitigation measures[10]. Perhaps even more challenging, the government should also promote the reform of China's land tenure system to facilitate large-scale farming[44]. Large-scale farms will be a better platform for the implementation of advanced management practices and mechanization (e.g., deep application of fertilizers) and can reduce the abatement cost per unit cropland area[10,25].

Livestock production is shifting from small-scale outdoor systems to large-scale intensive indoor systems[27], which causes decoupling between croplands for feed production and industrial feedlots[10]. As a consequence, manure is increasingly dumped or discharged instead of being recycled or reused owing to high transportation costs, resulting in huge $NH_3$ emissions in China[11]. In 2015 only 30% of livestock manure N was recycled to croplands in China[11] while in the EU the proportion was more than 65%[45]. Relocating feedlots to feed croplands can greatly improve manure recycling and reduce the associated implementation costs where livestock densities being kept within the cropland carrying capacity for manure application[11]. Financial incentives (e.g., subsidies, discounted interest, technical guidance, taxation exemption, etc.) are required to help farmers develop a region-specific farming structure that facilitates manure recycling, optimizes N management and promotes large-scale operation[27].

Furthermore, it should be noted that $NH_3$ mitigation through human dietary changes, also benefits human health[46,47]. Dietary change is a nontechnical measure with little implementation cost but requires other interventions to change consumers' preferences. The government can play an essential role in setting up campaigns to promote low-protein diets.

**Limitations and uncertainty**. Agricultural $NH_3$ mitigation strategies are linked to the overall N cycle and could affect agricultural production and farmers' incomes[7], which may further influence food security and rural economies. This study did not incorporate the effects of $NH_3$ mitigation on crop yield or animal productivity in the cost-benefit assessment of scenarios owing to the lack of comprehensive Chinese-specific data. In fact, fertilizer N application in China far exceeds the crop demand; $NH_3$

mitigation by improved farming practices would unlikely create N limitation or reduce crop yields[19,31]. If the yield benefits attributed to $NH_3$ mitigation could be quantified rationally and accurately, it would greatly improve the cost-effectiveness of $NH_3$ mitigation and therefore engage farmers to adopt these measures. Besides, this study does not address the regional difference in China due to the lack of detailed regional data. Given the large differences in regional agricultural structures and environmental conditions, mitigation strategies and targets may vary considerably, which affects the accuracy of current national estimates.

In this paper we limited the climate benefits to non-$CO_2$ GHG ($CH_4$ and $N_2O$) emissions resulting from $NH_3$ mitigation. This is mainly because $CO_2$ emission from agriculture is more related to fossil fuel consumption, such as fertilizer production and transportation[48], which is beyond the scope of this study. Furthermore, we did not consider the effects of $NH_3$ mitigation on climate change, including changes in aerosols and carbon sinks in terrestrial ecosystems, owing to limited research and models that target China[49,50].

There are complex chemical interactions among $SO_2$, $NO_x$, and $NH_3$ in the atmosphere[51]. Thus, the future policies to control $SO_2$ and $NO_x$ emissions may partially offset the effects of $NH_3$ mitigation, which also bring uncertainties to our estimations. While the projections of $NH_3$ mitigation potential and costs toward 2050 are based on current technologies, future technological advancement, and policy optimization may further reduce the implementation cost to yield a higher NEB. Nevertheless, as the first attempt, this study provides a basis and reference for ongoing improvement in $NH_3$ mitigation potential and cost-benefit assessment.

## Methods

**Selection of available mitigation options**. To identify feasible and cost-effective $NH_3$ abatement measures for Chinese agriculture, we reviewed all currently available mitigation options in China and other countries. Our criteria for the selection of abatement measures focus on five aspects:

(1) Mitigation efficiency: measures that could significantly reduce $NH_3$ emissions are included, for example, deep manure placement has a very high mitigation potential at 93–99%[52].

(2) Implementation cost: measures with lower cost or labor inputs are more acceptable to farmers, for example, reduced use of urea-based fertilizer and lower crude protein diet.

(3) Practical applicability: measures with current limited applicability due to technical, political or obvious social barriers in China, were excluded, for example, soil testing has been ruled out in this study due to high costs for the small farm size and high spatial and temporal variability, although it is an effective measure to optimize fertilizer use in the US and Europe where farm sizes are much larger.

(4) Limitations: measures that likely and significantly reduce agricultural productivity (crop yield or animal productivity) were adopted with caution, for example, the full substitution of synthetic fertilizers by manure decreases the yield of upland crops and lowland rice by 9.6% and 4.1%, respectively[53]; and low crude protein (LCP) feeding should only be adopted to an applicable level to avoid undermining animal productivity and welfare. Besides, LCP is mainly applicable to indoor animals (pig, poultry, and dairy).

(5) Presence of co-benefit: measures that could reduce both $NH_3$ emission and total GHG ($CH_4$ and $N_2O$) emissions are included, for example, biochar additives could reduce $NH_3$, $N_2O$ and $CH_4$ emissions during manure composting[54].

Based on the selection criteria and literature review, a total of 27 technical mitigation options for specific crops and animals were included in this study, with a coded version provided in Supplementary Tables 1 and 2. Detailed descriptions of these options are listed in Supplementary Tables 3–6 and Supplementary Notes 1–3. Most of these mitigation measures have been validated and adopted in the EU, while some of the measures (e.g., optimum N fertilization techniques) have been validated in China. For the measures that have been validated in China we directly adopted their parameters, whereas for measures that have not been validated in China, we calculated their potential implementation costs based on China-specific parameters such as labor cost, fertilizer prices, machinery cost. Only cost-effective measures and their combinations were selected for the analysis.

Most agricultural $NH_3$ and GHG emissions originate from the same activities (Supplementary Fig. 2) and their emission rates depend on common factors, such as management practice, weather conditions and soil type[7]. $NH_3$ abatement options can increase or decrease GHG emission[20]. This study aims to explore the maximum $NH_3$ mitigation potential while achieving the co-benefits of GHG reduction. In this context, optimal combinations of $NH_3$ mitigation options for different crops and animals are proposed in Supplementary Table 7 with their abatement efficiencies.

**Mitigation potential of $NH_3$ emissions.** $NH_3$ emissions from agriculture generally are assessed by multiplying the activity level with specific emission factors for each sector. The $NH_3$ mitigation potential is calculated as Eq. (1):

$$\Delta E_{NH_3} = \sum_i A_{i,k} * EF_{i,k} \times \eta_{i,k} \times X_{i,k}, \tag{1}$$

where $\Delta E_{NH_3}$ is the reduction of agricultural $NH_3$ emissions in mainland China; $i$ represents the source type; $k$ means a specific abatement option or the combination of multiple options; $A_{i,k}$ is the activity data of the source type; $EF_{i,k}$ is the original emission factor; $\eta_{i,k}$ is the $NH_3$ abatement efficacy; $X_{i,k}$ is the implementation rate of the abatement technique or options.

**Calculation of $NH_3$ abatement cost.** Abatement cost of $NH_3$ emissions in this study is defined as direct expenditures (the sum of investment costs and operation costs) for implementation of measures to reduce $NH_3$ emissions from agriculture, while the possible public costs (e.g., subsidy to promote the control policy) are not considered. Here, we mainly refer to the methodology of cost assessment from the GAINS model[55] to calculate the abatement costs of implementing various $NH_3$ mitigation measures. China-specific commodity prices were collected mainly from the *China Agricultural Products Cost-Benefit Yearbook* (2000–2018)[26], European cost data were adopted by conversion at market exchange rates where data supply is insufficient. All cost data from the literature were adjusted by the purchasing power parity (PPP) index and measured in constant 2015 US\$ (e.g., 100 EUR = US\$113.49, 100 CNY = US\$14.89) by assuming a 2% annual inflation and setting 2015 as the base year for future projection. The calculation of abatement costs is formulated in Eq. (2):

$$AC_{i,k} = I_{i,k} * \left[ \frac{(1+r)^{lt_{i,k}} \times r}{(1+r)^{lt_{i,k}} - 1} \right] + FO_{i,k} + VO_{i,k} - FS_{i,k}, \tag{2}$$

where $AC_{i,k}$ represents the annual implementation cost; $I_{i,k}$ refers to the investment cost; $r$ is the discount rate; $lt_{i,k}$ represents the lifetime of abatement technique (10–15 years); $FO_{i,k}$ is the fixed operating cost; $VO_{i,k}$ is the variable operating costs (e.g., feed, gas, electricity, labor, and water); $FS_{i,k}$ means saving costs from reduced use of N fertilizer.

Investment cost $I_{i,k}$ is calculated as a function of the average farm size ($AFS_i$) by Eq. (3):

$$I_{i,k} = ci_{i,k}^f \cdot st_i \cdot mp_i \cdot pc_i + \frac{ci_{i,k}^v}{AFS_i}, \tag{3}$$

where $ci^f$, $ci^v$ represents the fixed and variable coefficients derived from Klimont and Winiwarter (Annex: Table A1)[56]; $st_i$ represents manure storage time (in year); $mp_i$ represents manure production of a single animal per year; $pc_i$ represents production cycles per year; parameters used in the function are available in an online GAINS report.

Annual fixed operating costs $FO_{i,k}$ are estimated as the 0.05% of the total investments by Eq. (4) according to GAINS cost calculation[21].

$$FO_{i,k} = I_{i,k} \cdot 0.05\%. \tag{4}$$

Variable operating costs $VO_{i,k}$ are calculated by cost summation of the quantity ($Q$) of a certain extra supply (e.g. feed, gas, electricity, water, and labor) for a specific abatement option ($k$), as shown in Eq. (5):

$$VO_{i,k} = \sum_P \left( Q_{i,k} \cdot c_{i,k} \right), \tag{5}$$

where $p$ represents parameter type (additional feed, gas, electricity, water and labor input); $c_{i,k}$ means the unit price of these extra supply, which is mainly derived from the *China Agricultural Products Cost-Benefit Yearbook*[26] and market survey or adjusted by a coefficient if no direct data source could be accessed. The unit labor cost of farmworkers in 2015 is 15.7 Chinese yuan (CNY) per hour according to the national averaged salary for individual persons[26,57]. Other relevant parameters used in the calculation of FO and VO are obtained from GAINS.

The cost-effectiveness of various $NH_3$ mitigation options was calculated following Eq. (6)[55,58] to yield MACC curve according to increasing cost-effectiveness, as shown in Fig. 1.

$$CE_{i,k} = \frac{CE_{i,k} * \eta_{i,k} - CE_{i,k-1} * \eta_{i,k-1}}{\eta_{i,k} - \eta_{i,k-1}}, \tag{6}$$

where $CE_{i,k}$ is the cost-effectiveness for mitigation option $k$; $\eta_{i,k}$ is the $NH_3$ mitigation efficiency.

**Scenario analysis of future $NH_3$ emissions.** To explore the feasibility of $NH_3$ mitigation in the future, the CHANS model was employed in this study to make systematic and comprehensive analyses of $NH_3$ sources, emissions, and environmental fates[8]. A detailed introduction of the model can be found in Zhang et al.[8] and Gu et al.[59]. Taking into consideration the impacts of policy, and other external factors on Chinese agricultural production and consumption, the baseline $NH_3$ emission budgets during 2020–2050 were built in the first place, then four abatement scenarios (DIET, NUE, REC, and ALL) with corresponding packages of mitigation measures (detailed description in Table 3) were integrated into the CHANS model to quantify the new $NH_3$ emission budgets and identify the feasible $NH_3$ reduction potential in China. Human population and the per capita gross domestic product are two important parameters that affect future $NH_3$ emission budgets. These two parameters are assumed to remain the same as the BAU for the four mitigation scenarios while other input drivers and parameters, such as diet structure, NUE, cropping area, animal numbers, will vary with scenarios (Supplementary Fig. 6). Details about the data sources, prediction methods and parameters can be found in Supplementary Tables 8–18 and Supplementary Note 4. It should be noted that change in human diet structure as a nontechnical measure was also included in the scenario analysis to obtain a more comprehensive assessment of the mitigation potential and pathways.

**Societal benefit assessment of $NH_3$ mitigation.** Societal benefits ($SOC_{benefit}$) of $NH_3$ mitigation in this study are defined as the sum of benefits for human health ($HH_{benefit}$), ecosystem health ($EH_{benefit}$), GHG mitigation benefit ($GHG_{benefit}$) minus the cost of damage by increased acidity of precipitation ($AR_{damage}$, as shown in Eq. (7)

$$SOC_{benefit} = HH_{benefit} + EH_{benefit} + GHG_{benefit} - AR_{damage}. \tag{7}$$

The human health benefit assessment was performed based on the exposure–response function and the Value of Statistical Life (VSL) as applied in earlier studies both at the global and national scales[1]. Five categories of diseases causing premature mortality via $PM_{2.5}$ pollution are considered in this study, namely lower respiratory tract illness, chronic obstructive pulmonary disease (COPD), ischemic heart disease (IHD), lung cancer (LC) and cerebrovascular disease (CEV).

The impacts of $NH_3$ mitigation on annual $PM_{2.5}$ concentration were assessed based on the model simulation of Weather WRF-CMAQ performed by Xu et al.[60]. A deduced nonlinear function between $PM_{2.5}$ concentration and $NH_3$ reduction was built in Eq. (8); detailed description of WRF-CMAQ simulation can be found in Xu et al.[60] and Supplementary Note 5. Then, an exposure–response function (Eq. (9)) was combined with the health effect function (Eq. (10)) based on Global Burden of Disease[61] to estimate the 1-year premature mortality attributable to $PM_{2.5}$ exposure. Afterward, we used the updated Chinese-specific VSL following the method in Giannadaki et al.[1] to derive corresponding economic benefits of $NH_3$ abatement by Eq. (11) in China.

$$C_j = C_{2015} \times \left( 1 - 0.0173 \times e^{2.9532 \times \eta_j} \right), \tag{8}$$

$$HE_{j,q} = \sum_j e^{\beta_q \times (C_j - C_o)} \times HE_{0,q}, \tag{9}$$

$$\Delta M_j = \sum_q \left( HE_{j,q} - HE_{0,q} \right) \times Pop_j, \tag{10}$$

$$HH_{benefit,j} = VSL_j \times \Delta M_j, \tag{11}$$

where $C_j$ is the annual average $PM_{2.5}$ concentration in year $j$; $C_{2015}$ is the annual average $PM_{2.5}$ concentration in year 2015 ($50 \, \mu g \, m^{-3}$); $\eta_j$ is the reduction rate (%) of $NH_3$ emission; $q$ represents the category of diseases (IRL, COPD, IHD, LC, CEV); $\beta_q$ is the coefficient in the exposure–response function which refers to the proportion of change in the endpoint of each health effect of the population for unit change in $PM_{2.5}$ concentration; $C_0$ is the background concentration below which no health impact is assumed ($10 \, \mu g \, m^{-3}$ as suggested by the WHO[62]); $HE_{0,q}$ is the baseline health effect (the mortality risk) due to a particular disease category for China estimated by the WHO[61]; $HE_{j,q}$ is the actual health effect under significant $PM_{2.5}$ pollution levels; $Pop_j$ is the population exposed to air population in China; $\Delta M_j$ is the avoided death toll; $VSL_j$ is the Chinese-specific value of a statistical life derived from Giannadaki et al.[1]; $HH_{benefit,j}$ means the human health benefits by $NH_3$ mitigation.

Ecosystem benefits in this study are regarded as the avoided damage cost of decreased acidification and eutrophication of ecosystems due to $NH_3$ mitigation. We assume the unit $NH_3$ damage cost to the ecosystem in the European Nitrogen Assessment[21] is also applicable to China after correction for differences in the willingness to pay (WTP) for ecosystem service and PPP in China and EU, as

shown in Eq. (12).

$$EH_{benefit,j} = \Delta E_{NH_3,j} \times \partial_{EU} \times \frac{WTP_{China}}{WTP_{EU}} \times \frac{PPP_{China,j}}{PPP_{EU,j}}, \quad (12)$$

where $\partial_{EU}$ is the estimated unit ecosystem damage cost of $NH_3$ emission in relation to soil acidification and water eutrophication in Europe based on the European Nitrogen Assessment[63]; $WTP_{China}$ and $WTP_{EU}$ are the values of the WTP for ecosystem service in China and Europe; $PPP_{China,j}$ and $PPP_{EU,j}$ stand for the PPP of China and the EU.

GHG benefit from $NH_3$ mitigation is regarded as the avoided abatement cost of GHG emissions, as shown in Eqs. (13)–(14).

$$GHG_{benefit,j} = \Delta E_{GHG,j} * MCost_{GHG,j}, \quad (13)$$

$$\Delta E_{GHG,j} = \left( \Delta E_{directN_2O,j} + \Delta E_{indirectN_2O,j} \right) * 298 + \Delta E_{CH_4,j} * 34, \quad (14)$$

where $\Delta E_{GHG,j}$ is the estimated reduction in agricultural GHG emissions, presented as kg $CO_2$-eq, using the default values of 298 kg $CO_2$-eq for $N_2O$ emissions, and 34 kg $CO_2$-eq for $CH_4$ emissions at a 100-year time horizon[64]; both reduction of direct and indirect $N_2O$ emissions are included, the indirect $N_2O$ reduction is calculated as 1% of reduced $NH_3$ deposition according to the IPCC guideline[65]. $MCost_{GHG,j}$ represents the marginal abatement cost (the carbon price) to reduce one tonne of GHG emissions in $ per tonne $CO_2$-eq, the Chinese-specific (East Asia) value is derived from West et al.[66].

Acid rain damage ($AR_{damage,j}$) induced by $NH_3$ mitigation refers to the economic loss of reduced crop yields ($Crop_{damage,j}$) and forestry ($Forest_{damage,j}$) in Eq. (15). Based on the experimental results reported in Feng et al.[22] and model simulation of precipitation acidity in Liu et al.[21], we estimated the economic cost of increased acid rain under different mitigation scenarios.

$$AR_{damage,j} = Crop_{damage,j} + Forest_{damage,j}. \quad (15)$$

**Reporting summary**. Further information on research design is available in the Nature Research Reporting Summary linked to this article.

## Data availability

Data supporting the findings of this study are available within the article and its supplementary information files, or are available from the corresponding author upon reasonable request. Source data are provided with this paper.

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

## Acknowledgements
This study was supported by the National Key Research and Development Project of China (2018YFC0213300 and 2016YFC0207906), National Natural Science Foundation of China (41822701 and 41773068), Discovery Early Career Researcher Award of the Australian Research Council (DE170100423), the Melbourne Research Scholarship, Australia-China Joint Research Centre of Healthy Soils for Sustainable Food Production and Environmental Quality (ACSRF48165) and Linkage Projects of the Australian Research Council (LP160101417). We would like to thank Professor Wilfried Winiwarter, Arvin Mosier and Ian Willett in particular for their helpful comments on an earlier draft.

## Author contributions
B.G. and X.Z. designed the study. X.Z. performed the research. X.Z. and B.G. analyzed the data and interpreted the results. H.G. provided cost and benefit analysis supports. X.Z. and B.G. wrote the paper, D.C., S.L., X.L., and M.B. contributed to the discussion and revision of the paper.

## Competing interests
The authors declare no competing interests.
