## [Peer Review File · Nature Communications]

Reviewers' comments:

Reviewer #1 (Remarks to the Author):

I found this to be an interesting paper on the NH₃ mitigation potential and implementation costs for China. National studies of this nature are needed to inform future policy, and as China is an important emitter than this piece of work would form a novel and significant addition to the literature I would have liked to have seen a more joined up approach to this with GHG emissions and climate policies, but this nonetheless forms an important and relevant piece of work, that should find interest with a wide readership relating to land management, food systems, agricultural emissions and policy.

The main text well communicates the results and findings. I have a number of major and smaller comments that I recommend be addressed prior to acceptance for publication.

Major comments

1) On line 77 it is mentioned that many NH₃ abatement measures have not been validated for Chinese farming practices. It is not clear to me then how this issue is addressed by the authors. The efficacy of the abatement measures seem to be drawn from meta-analyses as described in Tables S1 and S2, some of which are China specific, and others global. How is the validity of these results for China's context decided on and appropriately adapted?

2) With reference to line 99, It was unclear to me how the included abatement measures were decided upon. How do you decide on what is acceptable for farmers (SI line 90), or which measures have limited applicability (SI line 91)? What data is used in support of these decisions? Can you talk through some instances that were ruled out on this basis?

3) With reference to the scenarios section around line 214, it isn't clear to me from the main text or SI how the input drivers (e.g. cropped area) have been decided in the scenario analyses. This is critical to understanding some of the core results (e.g. figures 2 and 3). For example, in the DIET scenario where animal production and feed production are reduced is any other type of production increased? What happens to land that was used for feed/grazed systems? It appears to me from Table S9 that production of almost all crops reduce? Would this externalise the emissions associated with food production? Has an established framework been used to construct the input drivers in the scenarios based on the trends and policies? Some studies are referenced on line 361, but it would be useful to communicate the approach.

4) Line 315: Can the authors further clarify if the effects of NH₃ mitigation on crop yield were incorporated in the scenarios. The yield changes here, but I am assuming this is due to changing cropping area only? Whilst there may not be the Chinese -specific data to estimate effects on yields or overall GHG emissions, could the authors give any insight into how important this might be based on other studies?

5) How valid are the methods used for quantifying the ecosystem benefits of avoided acidification and eutrophication given that they were developed for the European nitrogen assessment? Line 453

Smaller comments

6) Abstract, line 33: how much more mitigation potential does reduction in the consumption of animal products offer. Report the quantitative result in the abstract.

7) Table 1: Can the authors comment on why the damage cost estimate for China produced by this study is much narrower than estimates made in studies of the US and EU27?

8) Line 64: Statement refers to Fig S1 in the text, however I believe this is an error and should be Figure S2 (Agricultural NH₃ emissions related problems in China). This figure if it's to be included needs improving. The resolution is low and the graphics add little to the explanation. For me the statement in line 63 is enough to be understood without a figure. However, I would welcome the authors to comment on to what degree this issue is of importance to China. Decoupling of nutrient management between crop and animal systems is a globally widespread phenomena, but what is the scale of this issue in China compared to elsewhere?

9) Line 72/3 suggest deleting end of this sentence 'to improve both human...' to reduce repetition and improve readability.

10) Line 84-87 I would find it helpful as a reader if the structure of the results and discussion followed the same layout that is presented at the end of the introduction. Please adjust either the headings in the results/discussion, or the description of what is presented in lines 84-87 so that they more closely align.

11) Line 97: Who carries the abatement costs and does this matter for the recommendations?

12) Table 2: For the mitigation potential are the values in brackets ranges or percentiles? Some explanation in the table headings or the caption is needed.

13) Line 101: Space missing between fertilizer and (CRF) and irrigation and (SI..). These formatting errors likely introduced in track changes are common throughout the text. Please carefully edit the text to remove these issues. Other instances include lines 122, 209, 407, 418, 430, 431,

14) Line 114: 'introducing livestock unit' is an unusual phrase. LUs are an established concept and so they are not being introduced here.

15) Line 119: 'are given' rather than 'is showed'. As the LU coefficients are from another reference (Bai et al 2018), that should just be cited in the main text also. At present it makes it seem as if the authors have personally adapted LU coefficients for China.

16) Line 129: 'illustrates' rather than 'reveals'

17) Line 159: Suggest moving this closing statement to the top of the paragraph to improve readability and communication of the message.

18) Figure 3: spelling mistake in figure, 'implementation'.

Reviewer #2 (Remarks to the Author):

I recommend this manuscript be published after major reversion based on the following reasons.

(1) The authors declared they used WRF/CMAQ model to simulate the PM_{2.5} reduction by NH₃ mitigation, however, I cannot find any description for the modeling. The evaluations by comparing the simulations with measurements are necessary, because reliability of the human health cost estimation in this study depends strongly on the modeling results.

(2) China, not only emits a large amount of NH₃, but also SO₂ and NO_x, the acid rain is still a problem in China, especially in the southern China, Sichuan and Hunan etc. The acid rains could damage the crop yields and forest growth. The whole ecosystem health costs should include the acid rain effects.

(3) The future policy for air pollution control in China would still focus on SO₂ and NO_x. Their reductions could offset even distort the effects by NH₃ mitigation. The authors did not consider them in detail.

(4) It is interesting that the authors tell us that the farmers in China did not couple the farming and animal husbandry. If the authors discuss it in depth, e.g., the feasibility, cost and benefits, the comprehensive effects on human health and ecosystem health, the manuscript could be considered for publication.

Responses to Reviewers

Reviewer #1 (Remarks to the Author):

I found this to be an interesting paper on the NH₃ mitigation potential and implementation costs for China. National studies of this nature are needed to inform future policy, and as China is an important emitter than this piece of work would form a novel and significant addition to the literature I would have liked to have seen a more joined up approach to this with GHG emissions and climate policies, but this nonetheless forms an important and relevant piece of work, that should find interest with a wide readership relating to land management, food systems, agricultural emissions and policy. The main text well communicates the results and findings. I have a number of major and smaller comments that I recommend be addressed prior to acceptance for publication.

We appreciate the positive comments and feedback. We agree and have integrated NH₃ mitigation with agricultural GHG emissions in the revised MS.

We have added in the Introduction “*studies have also suggested that many NH₃ abatement techniques may simultaneously reduce agricultural methane (CH₄) and nitrous oxide (N₂O) emissions, bringing co-benefit for agricultural greenhouse gas (GHG) mitigations*” (Lines 64-66).

Based on previous experiments and meta-analyses (e.g. Hou et al., 2015; Xia et al., 2016; Wang et al., 2018; Luo et al., 2019), we selected effective and feasible measures for NH₃ and GHG mitigation different crops and animal types, as listed in updated SI Tables S6 and S7.

In the methods section, we have added the calculation methods of the GHG benefits:

“GHG co-benefit from NH₃ mitigation is regarded as the avoided abatement cost of GHG emissions, as showed in Eq. (9).

$$GHG_{benefit,j} = \Delta E_{GHG,j} * MCost_{GHG,j} \quad (9)$$

$$\Delta E_{GHG,j} = (\Delta E_{directN_2O,j} + \Delta E_{indirectN_2O,j}) * 298 + \Delta E_{CH_4,j} * 34 \quad (10)$$

where $\Delta E_{GHG,j}$ is the estimated reduction in agricultural GHG emissions, in kg CO₂-eq, using the default values of 298 kg CO₂-eq for N₂O emissions, and 34 kg CO₂-eq for CH₄ emissions at a 100-year time horizon; both reduction of direct and indirect N₂O emissions are included, the indirect N₂O reduction is calculated as 1% of reduced NH₃ deposition according to IPCC guidelines; $MCost_{GHG,j}$ represents the marginal abatement cost to reduce one tonne of GHG emissions in \$ per tonne CO₂-eq, the Chinese-specific (East Asia) value is derived from West et al. (2013).” (Lines 486-496)

In the Results and discussion section, we have added the co-benefits for GHG

mitigation: *“NH₃ mitigation could simultaneously reduce agricultural GHG emissions by 9-35% (101-385 Tg CO₂-eq) and by this generate climate benefits of US\$ 14-54 billion”* (Lines 127-129). The strong synergies of reducing co-emitted agricultural GHG emissions further improve the overall societal benefits of NH₃ mitigation and cost-efficiency of air pollution policies, supporting the implementation of climate policies.

We also briefly discussed the Limitation and Uncertainty related to GHG mitigation: *“In this paper, we limited the climate benefits to non-CO₂ GHG (CH₄ and N₂O) emissions resulting from NH₃ mitigation. This was mainly because CO₂ emission from agriculture is more closely related to fossil fuel consumption, such as fertilizer production and transportation, which is beyond the scope of this study that focuses on NH₃ emission during nutrient management”* (Lines 333-337).

Reference:

- (1) Hou, Y., Velthof, G.L. & Oenema, O. Mitigation of ammonia, nitrous oxide and methane emissions from manure management chains: a meta-analysis and integrated assessment. *Global Change Biol.* 21, 1293-1312 (2015).
- (2) Xia, L., et al. Can knowledge-based N management produce more staple grain with lower greenhouse gas emission and reactive nitrogen pollution? A meta-analysis. *Global Change Biol.* (2016).
- (3) Wang, Y., et al. Mitigating greenhouse gas and ammonia emissions from beef cattle feedlot production - A system meta-analysis. *Environ. Sci. Technol.* (2018).
- (4) Luo, Z., Lam, S.K., Fu, H., Hu, S. & Chen, D. Temporal and spatial evolution of nitrous oxide emissions in China: Assessment, strategy and recommendation. *J. Clean. Prod.* 223, 360-367 (2019).

Major comments

1) On line 77 it is mentioned that many NH₃ abatement measures have not been validated for Chinese farming practices. It is not clear to me then how this issue is addressed by the authors. The efficacy of the abatement measures seems to be drawn from meta-analyses as described in Tables S1 and S2, some of which are China specific, and others global. How is the validity of these results for China's context decided on and appropriately adapted?

We appreciate the reviewer's concern. For the measures which have been validated in other countries but not in China, we assume their reduction efficiencies can be derived from previous global meta-analysis (e.g. Ti et al., 2019; Pan et al., 2016) for application to China. We acknowledge that all these mitigation options have their ceiling in terms of mitigation efficiency, and always come with an uncertainty range (lower limits and upper limit value) of mitigation efficiency. Considering the issues of overfertilization, surface spreading and poor manure management in China, these mitigation measures would also be effective. To inform readers about the reliability and validity, we have included the uncertainty range of mitigation efficiency of measures' mitigation efficiency for different crops and animals in **Tables S1 and S2**. We also calculated their potential implementation costs based on China-specific parameters such as labour cost, fertilizer prices, machinery cost. These results were then included to assess the national NH₃ mitigation potential, presented in **Table 2**.

We have modified the sentence to read “Many NH₃ abatement measures have not been widely practiced in China and their implementation costs and the impact on agricultural GHG emissions have not been assessed” (Lines 72-74).

Reference:

- (1) Ti, C., Xia, L., Chang, S.X. & Yan, X. Potential for mitigating global agricultural ammonia emission: A meta-analysis. *Environ. Pollut.* 245, 141-148 (2019).
- (2) Pan, B., Lam, S.K., Mosier, A., Luo, Y. & Chen, D. Ammonia volatilization from synthetic fertilizers and its mitigation strategies: A global synthesis. *Agriculture, Ecosystems & Environment* 232, 283-289 (2016).

2) With reference to line 99, It was unclear to me how the included abatement measures were decided upon. How do you decide on what is acceptable for farmers (SI line 90), or which measures have limited applicability (SI line 91)? What data is used in support of these decisions? Can you talk through some instances that were ruled out on this basis?

Apologize for not clearly describing this. We have updated the Method section as follows: “To identify the feasible and cost-effective NH₃ abatement measures for Chinese agriculture, we reviewed all currently available mitigation options in China and other countries. Our criteria for the selection of abatement measures are: 1) mitigation efficiency, 2) implementation cost, 3) practical applicability 4) limitations, and 5) presence of co-benefits” (Lines 350-354). Details of the selection criteria are given in SI methods (SI Lines 95-115). Based on these selection criteria a total of 27 technical mitigation options for specific crops and animals were included in this study (Tables S1 and S2).

Unlike industrialized, high income countries, which generally have larger farm sizes and higher levels of mechanization, farms in China are mostly small (less than 1 hectare) and farmers are typically resource-limited and knowledge-poor. Engaging millions of Chinese smallholder farmers to adopt the environmentally friendly management practices is challenging (Cui et al., 2018). To incentivize their adoption, the selected measures should be cheap, profitable, and easy to operate.

To support the selection of feasible measures for China, datasets as provided by Reis et al., (2015; Costs of Ammonia Abatement and the Climate Co-Benefits), in the China Agricultural Products Cost-Benefit Yearbook (NDRC, 2000-2018) and in Guidelines for major crop fertilization in China (Zhang et al., 2009) have been used to specify the practical aspects of adoption of NH₃ mitigation options applied to cropland and animal production in China (Tables S3 and S4). Further, national policies and plans (e.g. Zero Increase Action Plan on Fertilizer Use by 2020, Recommendations on accelerating the resource utilization of livestock waste, and National sustainable agricultural development plan (2015-2030)) have been used to determine the parameters and implementation levels of different options. Chinese current farming practice and future opportunities of the selected measures were included in Table S5. MACC analysis was then used to prioritize these mitigation options based on their cost-efficiency (Fig 1).

For illustration we give an example, “Owing to the high cost for small farms and high spatial variability soil testing has been excluded in this study although it is effective in optimizing fertilizer use in the US and Europe, where farms are generally much larger” was added (SI Lines 103-106).

Reference:

- (1) Cui, Z., et al. Pursuing sustainable productivity with millions of smallholder farmers. *Nature* 555, 363-366 (2018).
- (2) Reis, S., Howard, C. & Sutton, M.A. *Costs of Ammonia Abatement and the Climate Co-Benefits* (Springer, 2015).
- (3) NDRC. *China Agricultural Products Cost-Benefit Yearbook (2000-2018)*. (China Statistics Press, 2019).
- (4) Zhang, F., Chen, X. & Chen, Q. *Guidelines for major crop fertilization in China* (China Agriculture Press, 2009).

3) With reference to the scenarios section around line 214, it isn't clear to me from the main text or SI how the input drivers (e.g. cropped area) have been decided in the scenario analyses. This is critical to understanding some of the core results (e.g. figures 2 and 3). For example, in the DIET scenario where animal production and feed production are reduced is any other type of production increased? What happens to land that was used for feed/grazed systems? It appears to me from Table S9 that production of almost all crops reduce? Would this externalise the emissions associated with food production? Has an established framework been used to construct the input drivers in the scenarios based on the trends and policies? Some studies are referenced on line 361, but it would be useful to communicate the approach.

We have added the details about how the input drivers were decided in revised SI as follows: “The framework of input drivers used for the scenario analyses are summarized in Fig S6 (a new figure), which is based on the N balance principle i.e. future food N supply from crops and livestock production should meet the demands of human consumption. We first projected the future human population, GDP, diet preference and urbanization to quantify future food consumption; then the demands of crop and animal food were estimated; third, the required crop and livestock production, cropping area, fertilizer use and manure production were calculated for subsequent scenario analysis.” (SI Lines 390-395)

REC and NUE scenarios are assumed to have the same food consumption with BAU scenario, while ALL scenario has the same food consumption with DIET scenario, but different fertilizer use and manure management practices. The proposed DIET scenario reduces the human consumption of livestock products while increasing that of crop and vegetable production, as compared to the BAU scenario. We have added a new Table (Table S10) to explain the change of food consumption.

Because the land area released from the reduction of growing animal feed is much larger, the net land requirement for crop and livestock production under DIET scenario declines (see updated Table S12), which will alleviate the tensions between urban growth and agricultural land use and preservation (Jiang et al., 2013). The reduced livestock production and subsequent feed production under the DIET scenario therefore will decrease livestock manure and fertilizer use,

then externalise reduced NH₃ emission.

Reference:

- (1) Jiang, L., Deng, X. & Seto, K.C. The impact of urban expansion on agricultural land use intensity in China. *Land Use Policy* 35, 33-39 (2013).

Table S10 Food consumption under BAU and DIET scenario

Food consumption (kg/y/cap)	2015	2020	2025	2030	2035	2040	2045	2050
BAU-grain	127	120	115	110	105	100	95	90
BAU-fruits	85	100	105	110	115	120	125	130
BAU-vegetables	98	100	107	114	121	128	135	142
BAU-beans	8	8	8	8	8	8	8	8
BAU-livestock meat	35	40	45	50	55	60	65	70
BAU-fish	11	12	15	18	21	24	27	30
BAU-eggs	10	11	12	13	14	15	16	17
BAU-milk	12	15	25	35	45	55	65	75
DIET-grain	127	118	118	118	118	118	118	118
DIET-fruits	85	100	102	104	106	108	110	112
DIET-vegetables	98	100	105	110	115	120	125	130
DIET-beans	8	9	9	9	10	10	11	11
DIET-livestock meat	35	40	40	39	39	38	38	37
DIET-fish	11	11	12	13	14	15	16	17
DIET-eggs	10	11	12	12	13	13	14	14
DIET-milk	12	12	22	32	42	52	62	72

Note: the prediction of food consumption under BAU scenario is based on current high-income countries diet structure (animal food N ratio=60%); the prediction of food consumption under DIET scenario is based on the Dietary guidelines for Chinese residents (animal food N ration=40%).

Figure S6. The framework of input drivers in scenario analysis (2020-2050)

Note: White rectangles with blue edges represent the basic parameters, arrows stand for data flow, green rounded rectangles represent different scenarios, while orange rounded rectangles with red edges represent the corresponding agricultural activity output.

4) Line 315: Can the authors further clarify if the effects of NH_3 mitigation on crop yield were incorporated in the scenarios. The yield changes here, but I am assuming this is due to changing cropping area only? Whilst there may not be the Chinese-specific data to estimate effects on yields or overall GHG emissions, could the authors give any insight into how important this might be based on other studies?

The effects of NH_3 mitigation on crop yield were not incorporated in the mitigation scenarios owing to the lack of comprehensive and reliable Chinese-specific experimental data. In this study, we assume that four mitigation scenarios have the same crop yield with BAU scenario, we have added a new figure in SI (Fig. S7). The prediction of crop yield changes during 2020-2050 is based on previous studies (Ye et al. (2013); Ray et al., (2013)) and national plans to realize high-yield and efficient grain crop production. (e.g. National sustainable agricultural development plan (2015-2030))

Figure S7 Crop yield trends in China during 2020-2050

NH_3 mitigation in principle could increase crop yield by increasing N availability and N use efficiency (NUE) for crop growth. As current N fertilizer application in China far exceeds the crop demand, optimizing N application would unlikely create N limitation or reduce crop yields. We have added brief discussion related to crop yield in the Limitations and uncertainty section as follow: *“This study did not incorporate the effects of NH_3 mitigation on crop yield or animal productivity in the cost-benefit assessment of scenarios owing to lack of comprehensive Chinese-specific data. In fact, fertilizer N application in China far exceeds the crop demand; NH_3 mitigation by improved farming practices would unlikely create N limitation or reduce crop yields (Xia et al., 2017; Ju et al., 2009). If the yield benefits attributed to NH_3 mitigation could be quantified rationally and*

accurately, it would greatly improve the cost-effectiveness of NH₃ mitigation and therefore engage farmers to adopt these measures.” (Line 321-328). Further study is needed to bridge the knowledge gap in China.

The effect on agricultural GHG emissions due to NH₃ mitigation has been addressed earlier in response to the reviewer’s general comment.

Reference:

- (1) Ye, L., et al. Climate change impact on China food security in 2050. *Agron. Sustain. Dev.* 33, 363-374 (2013)
- (2) Ray, D.K., Mueller, N.D., West, P.C. & Foley, J.A. Yield Trends Are Insufficient to Double Global Crop Production by 2050. *PLoS One* 8, e66428 (2013).
- (3) Xia, L., et al. Can knowledge-based N management produce more staple grain with lower greenhouse gas emission and reactive nitrogen pollution? A meta-analysis. *Global Change Biol.* (2016).
- (4) Ju, X.T., et al. Reducing environmental risk by improving N management in intensive Chinese agricultural systems. *Proc. Natl Acad. Sci. USA* 106, 3041-3046 (2009).

5) How valid are the methods used for quantifying the ecosystem benefits of avoided acidification and eutrophication given that they were developed for the European nitrogen assessment? Line 453

We validated and estimated the ecosystem benefits (including avoided acidification and eutrophication) attributable to NH₃ mitigation for China based on the studies from the European Nitrogen Assessment (ENA, 2011), which is particularly reliant on a GDP dependent ‘willingness to pay’ (WTP) to prevent loss of biodiversity and ecosystem services due to NH₃ pollution. We added description in the Method section *“We assume the unit NH₃ damage cost to the ecosystem in the European Nitrogen Assessment is also applicable to China after correction for differences in the willingness to pay (WTP) for ecosystem services and Purchasing Power Parity (PPP) in China and EU, as shown in Eq. (8).*

$$EH_{benefit,j} = \Delta E_{NH_3,j} * \partial_{EU} * \frac{WTP_{China}}{WTP_{EU}} * \frac{PPP_{China,j}}{PPP_{EU,j}} \quad (8)$$

where ∂_{EU} is the estimated unit ecosystem damage cost of NH₃ emission in relation to soil acidification and water eutrophication in Europe based on the European Nitrogen Assessment; WTP_{China} and WTP_{EU} are value of the willingness to pay (WTP) for ecosystem services in China and EU; $PPP_{China,j}$ and $PPP_{EU,j}$ stand for the PPP of China and the EU” (Lines 474-484)

Although WTP to prevent ecosystem damage depends also on other parameters than income (e.g. culture, education), we believe it is the best approach currently available. Future studies in China are needed for more accurate estimation of the ecosystem damage costs of NH₃ emissions.

Reference:

- (1) Brink, C. & Van Grinsven, H. Costs and benefits of nitrogen in the environment The European Nitrogen Assessment: Sources, Effects and Policy Perspectives. (Cambridge University Press, 2011).
- (2) Van Grinsven, H.J.M., et al. Costs and Benefits of Nitrogen for Europe and

Implications for Mitigation. Environ. Sci. Technol. 47, 3571-3579 (2013).

Smaller comments

6) Abstract, line 33: how much more mitigation potential does reduction in the consumption of animal products offer. Report the quantitative result in the abstract.

We have added the number in abstract as suggested: "*Adoption of diets with less animal products offers a further NH₃ reduction potential of 12% by 2050*" (Lines 32-33).

7) Table 1: Can the authors comment on why the damage cost estimate for China produced by this study is much narrower than estimates made in studies of the US and EU27?

Previously we used damage cost of NH₃ emission in China adopted directly from Gu et al. (2015). In the revised MS, we updated the damage cost based on the European damage cost with conversion of Value of statistical life (VSL) of China and the EU from a recent study of Giannadaki et al. (2018). The value of health damage cost by NH₃ emission in China was re-estimated at US \$ 44-115 billion, similar to the range of US\$ 55-114 billion in the EU (Van Grinsven et al., 2018) and US\$ 69-180 billion in the US (Table 1). The damage cost of NH₃ emissions in the US was also updated based on a more recent study by Goodkind et al. (2019).

Reference:

- (1) Goodkind, A.L., Tessum, C.W., Coggins, J.S., Hill, J.D. & Marshall, J.D. Fine-scale damage estimates of particulate matter air pollution reveal opportunities for location-specific mitigation of emissions. Proc. Natl Acad. Sci. USA 116, 8775-8780 (2019).
- (2) Gu, B., Ju, X., Chang, J., Ge, Y. & Vitousek, P.M. Integrated reactive nitrogen budgets and future trends in China. Proc. Natl Acad. Sci. USA 112, 8792-8797 (2015).
- (3) Giannadaki, D., Giannakis, E., Pozzer, A. & Lelieveld, J. Estimating health and economic benefits of reductions in air pollution from agriculture. Sci. Total Environ. 622-623, 1304-1316 (2018).
- (4) Van Grinsven, H.J.M., et al. Reducing external costs of nitrogen pollution by relocation of pig production between regions in the European Union. Reg. Environ. Change 18, 2403-2415 (2018).

8) Line 64: Statement refers to Fig S1 in the text, however I believe this is an error and should be Figure S2 (Agricultural NH₃ emissions related problems in China). This figure if it's to be included needs improving. The resolution is low and the graphics add little to the explanation. For me the statement in line 63 is enough to be understood without a figure. However, I would welcome the authors to comment on to what degree this issue is of importance to China. Decoupling of nutrient management between crop and animal systems is a globally widespread phenomena, but what is the scale of this issue in China compared to elsewhere?

Apologies for this error. We agree and have removed this figure.

Both the largest livestock (and manure) production and greatest fertilizer use in the world occur in China. In 2015, only 30% of livestock manure N was recycled to croplands in China (Zhang et al.,2019) while in the EU this share was more than 65% (Bai et al., 2016). N loss from over-fertilization and manure management represents a substantial financial cost to farmers and worse still, it further causes substantial contamination of air, water, and soil (Zhang et al., 2019).

With the increasing demand for livestock products from population growth and dietary changes, more manure will be generated in the next 20 years, especially in peri-urban areas with concentrated animal feeding operations and intensive farms. To reduce N loss and the associated environmental pollution, rebuilding the linkage between livestock and cropland by manure recycling is urgently needed (Zhang et al., 2019).

We have added discussion on the feasibility (Lines 204-207), abatement cost (Lines 210-212) and societal benefits (Lines 232-237 and Fig. 3) of integration of crop and livestock production in the revised MS.

Reference:

- (1) Bai, Z., et al. Nitrogen, Phosphorus, and Potassium Flows through the Manure Management Chain in China. Environ. Sci. Technol. (2016).
- (2) Bai, Z., et al. China's livestock transition: Driving forces, impacts, and consequences. Science advances 4, r8534 (2018).
- (3) Zhang, C., et al. Rebuilding the linkage between livestock and cropland to mitigate agricultural pollution in China. Resources, Conservation and Recycling 144, 65-73 (2019).

9) Line 72/3 suggest deleting end of this sentence ‘to improve both human...’ to reduce repetition and improve readability.

Revised as suggested (Lines 77-79).

10) Line 84-87 I would find it helpful as a reader if the structure of the results and discussion followed the same layout that is presented at the end of the introduction. Please adjust either the headings in the results/discussion, or the description of what is presented in lines 84-87 so that they more closely align.

Thank you for the suggestion. We have adjusted both the Introduction and the headings of the Results and discussion section.

Introduction:

“1) to identify feasible NH₃ abatement options and estimate the agricultural NH₃ mitigation potential and the associated implementation costs and societal benefits, 2) to determine the mitigation priority and strategies for China, and 3) to explore optimal NH₃ mitigation pathways for the next 30 years (2020-2050) by scenario analysis and cost-benefit assessment.” (Line 81-85)

Results and discussion:

“NH₃ mitigation potential, abatement costs and societal benefits. (Line 89)
NH₃ mitigation priority and strategies for China. (Line 136)
NH₃ mitigation pathways in 2020-2050.” (Line 173)

11) Line 97: Who carries the abatement costs and does this matter for the recommendations?

Ideally all beneficiaries of Chinese agriculture should, directly or indirectly, pay their fair share of the NH₃ abatement costs. In the current Chinese situation, government (and public resources) and the farmers are the stakeholders bearing the cost of NH₃ abatement, while agri-food industry and retailers are the intermediaries between farmers and consumers, and should also facilitate the transfer of abatement costs. Eventually, this could lead to fairer sharing of mitigation, where the consumers of products with high NH₃ losses pay more. However, we believe this discussion on principles and policies of burden-sharing is beyond the scope of this paper.

Since small farmers dominates in China, strategic design for cost-effective mitigation pathways is needed. Therefore “cheap and easy mitigation options (direct reduction of N fertilizer use and improved animal feeding practices) should be introduced first to pick the “low-hanging fruit” of NH₃ mitigation in China. The remaining mitigation measures (e.g. housing adaptation and manure handling systems) are expensive due to the higher requirements of the investments in technologies and infrastructures. It is necessary to increase government support (e.g. technical guidance and training) and subsidies (e.g. enhanced efficiency fertilizers and agricultural machinery) to encourage farmers to adopt these mitigation measures” (Line 289-296)

12) Table 2: For the mitigation potential are the values in brackets ranges or percentiles? Some explanation in the table headings or the caption is needed.

The values in brackets represent the uncertainty range of mitigation potential with 95% CI. We have updated the **Table 2** to show only the potential range. For example, the previous mitigation potential for cropland was “64% (39-68%)”, it has been changed to “39-68%”.

13) Line 101: Space missing between fertilizer and (CRF) and irrigation and (SI..). These formatting errors likely introduced in track changes are common throughout the text. Please carefully edit the text to remove these issues. Other instances include lines 122, 209, 407, 418, 430, 431,

Revised as suggested.

14) Line 114: ‘introducing livestock unit’ is an unusual phrase. LUs are an established concept and so they are not being introduced here.

We have modified this sentence as “This study compares the unit abatement costs among different animal types (Table 2) after conversion to livestock unit (LU)” (Lines 107-108)

15) Line 119: 'are given' rather than 'is showed'. As the LU coefficients are from another reference (Bai et al 2018), that should just be cited in the main text also. At present it makes it seem as if the authors have personally adapted LU coefficients for China.

Revised as suggested. (Line 110)

16) Line 129: 'illustrates' rather than 'reveals'

Revised as suggested (Line 140)

17) Line 159: Suggest moving this closing statement to the top of the paragraph to improve readability and communication of the message.

Revised as suggested (Lines 150-152).

18) Figure 3: spelling mistake in figure, 'implementation'.

Revised as suggested (Fig.3)

Reviewer #2 (Remarks to the Author):

I recommend this manuscript be published after major revision based on the following reasons.

We thank the reviewer for his/her comments.

1) The authors declared they used WRF/CMAQ model to simulate the PM_{2.5} reduction by NH₃ mitigation, however, I cannot find any description for the modelling. The evaluations by comparing the simulations with measurements are necessary, because reliability of the human health cost estimation in this study depends strongly on the modelling results.

Apologize for not clearly describing the analysis. We have added description of the WRF/CMAQ in the SI as follow: "*A detailed sensitivity simulation of PM_{2.5} reduction by NH₃ mitigation using the WRF-CMAQ model in China was conducted by Xu et al (2017). When NH₃ emission is cut by 20, 40, 60, 80 and 100%, the concentration of PM_{2.5} would decline by 2.7, 6.3, 11.3, 19.0 and 29.8%, respectively (Fig. S11, a new figure). Based on the results from WRF-CMAQ simulation, a nonlinear function (Eq. (4), $R^2=0.9839$) between PM_{2.5} concentration and NH₃ reduction was deduced which was used for the scenario analysis in this study*" (SI Line 442-447)

Figure S11 The relationship between $PM_{2.5}$ concentration and NH_3 mitigation based on WRF-CMAQ simulation.

$$C_j = C_{2015} * (1 - 0.0173 * e^{2.9532 * \eta_j}) \quad (4)$$

“where C_j is the annual average $PM_{2.5}$ concentration in year j ; C_{2015} is the annual average $PM_{2.5}$ concentration in year 2015 ($50 \mu g m^{-3}$); η_j is the reduction rate (%) of NH_3 emission” (Lines 461-462)

In this study, we assumed that SO_2 and NO_x will remain at the same emission control level during WRF/CMAQ simulation in Xu et al (2017). When NH_3 emission is cut by 50%, the associated $PM_{2.5}$ concentrations would decline by 7.6%, which could reduce the premature mortalities by approximately 6-16%, and is generally consistent with previous studies (Giannadaki et al (2018); Hu et al.(2017)). Table S23 gives the reduction of premature mortalities under future NH_3 mitigation scenarios” (SI Lines 448-453).

Table S22 Reduction of premature mortalities under NH_3 mitigation scenarios

	Premature mortalities	Reduction of premature mortalities			
	BAU (million)	DIET	NUE	REC	ALL
2020	1.03	4%	4%	3%	7%
2025	1.12	5%	5%	4%	10%
2030	1.20	5%	5%	4%	13%
2035	1.23	6%	6%	4%	15%
2040	1.23	6%	6%	5%	18%
2045	1.23	6%	6%	6%	20%
2050	1.22	6%	7%	7%	21%

The main description and validation of WRF-CMAQ simulation in Xu et al (2017) are listed as follow:

“Simulation period: January, April, July and October of 2015, and the time interval of the output is 1h.

Simulation area: the CMAQ model adopts the Lambert projection coordinate

system, the central longitude is 103°E, the central latitude is 37°N, the two parallel latitudes are 25°N and 40°N, respectively. The horizontal simulation range for X direction is -2690 to 2690 km, for Y direction is -2150 to 2150 km with grid spacing of 20 km. The whole China is divided into 270×216 grids. A total of 14 pressure layers were set in the vertical direction, and the layer spacing gradually increased from bottom to top.

Meteorological simulation: the meteorological field required by the CMAQ model is provided by the mesoscale meteorological model WRF. The WRF model and the CMAQ model adopt the same simulation period and spatial projection coordinate system, but the simulation range of WRF is larger than that of CMAQ. The horizontal simulation range for X direction is -3600 km to 3600 km, for Y direction is -2520 km to 2520 km, with the grid spacing of 20km.

Parameterization scheme of WRF-CMAQ was summarized in Table S22, as shown below:

Table S22 Parameterization scheme of WRF-CMAQ

Parameterization scheme	CMAQ
Version	5.0.2
Grid nesting	Single-layer grid
Horizontal resolution	20 km
Vertical layers	14
Gas phase chemistry	CB05
Aerosol chemistry	AERO5
Photochemical rate	In-line
The wind dust	Off
The boundary conditions	Default
The initial conditions	Restart every single day
Parameterization scheme	WRF
Version	3.4
Microphysical schemes	WSM6
Longwave Radiation	New Goddard scheme
Shortwave Radiation	RRTM
Surface layer	Pleim Xiu
Interaction with Earth's surface	Pleim Xiu
Boundary layer	ACM2
Cumulus convection	Kain-Fritsch

Model validation: The monthly CMAQ model simulation results were verified with the actual monthly observation data of 302 cities where PM_{2.5} monitoring was conducted in 2015 (Fig. S9). The results show that CMAQ model simulation generally agrees with the observations; the correlation coefficient between observed and simulated annual average PM_{2.5} concentration was 0.82 (n=302, p<0.05), with normalized mean biases (NMB) of -21.67 and Normalized mean error (NME) of 29.49.”(SI Lines 456-476)

[REDACTED]

Reference:

- (1) Xu, Y., et al. Sensitivity analysis of PM_{2.5} pollution to ammonia emission control in China. *China Environmental Science* 37, 2482-2491 (2017).
- (2) Giannadaki, D., Giannakis, E., Pozzer, A. & Lelieveld, J. Estimating health and economic benefits of reductions in air pollution from agriculture. *Sci. Total Environ.* 622-623, 1304-1316 (2018).
- (3) Hu, J., et al. Premature Mortality Attributable to Particulate Matter in China: Source Contributions and Responses to Reductions. *Environ. Sci. Technol.* 51, 9950-9959 (2017).

2) China, not only emits a large amount of NH₃, but also SO₂ and NO_x, the acid rain is still a problem in China, especially in the southern China, Sichuan and Hunan etc. The acid rains could damage the crop yields and forest growth. The whole ecosystem health costs should include the acid rain effects.

Thanks for the insightful comments. We agree and have included the damage cost of acid rain in the assessment of overall societal benefit analysis in the Method section as:

“Societal benefits ($S_{benefit}$) of NH₃ mitigation in this study are defined as the sum of benefits for human health ($HH_{benefit}$), ecosystem health ($EH_{benefit}$), GHG co-benefit ($GHG_{benefit}$) minus the cost of damage by increased acidity of precipitation (AR_{damage}), as shown in Eq. (3)” (Lines 436-440).

$$S_{benefit} = HH_{benefit} + EH_{benefit} + GHG_{benefit} - AR_{damage} \quad (3)$$

Recent study (Liu et al., 2019) suggested that a 50% reduction of NH₃ emission

could decrease rainfall pH by 1.0 unit and substantially increase the areas with heavy acid rain, resulting in large potential economic loss due to reduced crop yields and forestry. To assess the economic loss of acid rain, we have added “Based on the experimental results reported in Feng et al. (2002) and model simulation of precipitation acidity in Liu et al. (2019), we estimated the damage cost of increased acid rain under different NH₃ mitigation scenarios” (Lines 498-501).

We have added in the Result and discussion section “However, reduction of NH₃ might increase acidity of precipitation and cause an economic loss of US\$ 4-7 billion” (Lines 129-130). And “The reduction of agricultural NH₃ emissions may aggravate acid rain over China and may result in economic loss under all mitigation scenarios (Fig. 3). Nevertheless, the overall societal benefits under these scenarios still exceed the corresponding abatement costs and result in net national social welfare” (Lines 234-237).

Reference:

- (1) Liu, M., et al. Ammonia emission control in China would mitigate haze pollution and nitrogen deposition but worsen acid rain. Proc. Natl Acad. Sci. USA, 201814880 (2019).
- (2) Feng, Z., Miao, H., Zhang, F. & Huang, Y. Effects of acid deposition on terrestrial ecosystems and their rehabilitation strategies in China. J. Environ. Sci.-China 14, 227-233 (2002).
- 3) The future policy for air pollution control in China would still focus on SO₂ and NO_x. Their reductions could offset even distort the effects by NH₃ mitigation. The authors did not consider them in detail.

Thanks for the insightful comments. We have added analysis and discussion on the interactions among SO₂ and NO_x control and NH₃ mitigation following your suggestions. The new section reads as follow:

“Although the continuing stringent policies to reduce SO₂ and NO_x emission could further improve air quality and may partially offset the effects of NH₃ mitigation, studies have suggested that the policies are not sufficient or cost-effective in achieving the targets of clean air in China (Wu et al., 2016; Bai et al., 2019; Zhang et al., 2020). A recent study has found that reducing livestock NH₃ emissions would be highly effective in reducing PM_{2.5} during severe winter haze events (Xu et al., 2019). Our quantitative assessments of the implementation cost and societal effects of NH₃ mitigation in China further demonstrate that NH₃ mitigation could generate net societal benefits, even though it may worsen regional acid rain. Therefore, coordinated mitigation of multi air pollutants (SO₂, NO_x and NH₃) should be implemented to more rationally and effectively achieve the dual benefits of protecting human and ecosystem health in China at both national and regional scales (Liu et al., 2019).” (Lines 277-287).

Reference:

- (1) Wu, Y., et al. PM_{2.5} pollution is substantially affected by ammonia emissions in China. Environ. Pollut. 218, 86-94 (2016).

- (2) Bai, Z., et al. Further Improvement of Air Quality in China Needs Clear Ammonia Mitigation Target. *Environ. Sci. Technol.* 53, 10542-10544 (2019).
- (3) Zhang, F., et al. An unexpected catalyst dominates formation and radiative forcing of regional haze. *Proc. Natl Acad. Sci. USA* 117, 3960-3966 (2020).
- (4) Xu, Z., et al. High efficiency of livestock ammonia emission controls in alleviating particulate nitrate during a severe winter haze episode in northern China. *Atmos. Chem. Phys.* 19, 5605-5613 (2019).
- (5) Liu, M., et al. Ammonia emission control in China would mitigate haze pollution and nitrogen deposition but worsen acid rain. *Proc. Natl Acad. Sci. USA*, 201814880 (2019).

4) It is interesting that the authors tell us that the farmers in China did not couple the farming and animal husbandry. If the authors discuss it in depth, e.g., the feasibility, cost and benefits, the comprehensive effects on human health and ecosystem health, the manuscript could be considered for publication.

We appreciate this suggestion and have added more discussion in the revised MS. **Feasibility:** “The total excretion nitrogen (N) generated by livestock production was 13.4 Tg N in 2015 and is estimated to reach 18.2 Tg N in 2050 under REC scenario. Meanwhile, the total N demand of crops in China is estimated to be 25.0 Tg N in 2050, suggesting that animal excretion N would be within the cropland carrying capacity in China” (Lines 204-207).

Abatement cost: “Under the assumption that 60% manure N is recycled to croplands, the REC scenario could save 10.9 Tg chemical fertilizer N use and reduce NH₃ emission by 24% (3.3 Tg N) in 2050. The abatement cost is estimated at US\$ 3.9 billion acknowledging the considerable socio-economic barriers of relocation and adaptation of the livestock supply chain and transport infrastructure” (Lines 208-212).

Societal benefit: The corresponding societal benefits for human health and ecosystem health are estimated at US\$ 16.1 and 8.5 billion, respectively (Fig.3). REC could also simultaneously reduce GHG emission of 109 Tg CO₂-eq and generate climate benefits of US\$ 10.3 billion. Although it may increase damage of acid rain at a damage cost of US\$ 2.7 billion, the overall societal benefits still far exceed the abatement costs (Fig.3). Therefore, coupling crop and livestock production with manure nutrient recycling is beneficial for both human health and ecosystem health without substantial costs.

Figure 3 Costs and benefits of NH₃ mitigation during 2020-2050 in China.

We have modified **Policy implications as follow:** “Livestock production is shifting from small-scale outdoor systems to large-scale industrial indoor systems, which causes decoupling between croplands for feed production and industrial feedlots. As a consequence, manure is increasingly dumped or discharged instead of being recycled or reused owing to high transportation costs, resulting in huge NH₃ emissions in China. In 2015, only 30% of livestock manure N was recycled to croplands in China (Zhang et al.,2019) while in the EU the proportion was more than 65% (Bai et al., 2016). Relocating feedlots to feed croplands can greatly improve the potential for manure recycling and reduce the associated implementation cost where livestock densities being kept within the cropland carrying capacity for manure application. Financial incentives (e.g. subsidies, discounted interest, technical guidance, taxation exemption etc.) are required to help farmers develop a region-specific farming structure that facilitates manure recycling, optimizes N management and promotes large-scale operation.” (Lines 301-312)

Reference:

- (1) Bai, Z., et al. Nitrogen, Phosphorus, and Potassium Flows through the Manure Management Chain in China. Environ. Sci. Technol. (2016).
- (2) Zhang, C., et al. Rebuilding the linkage between livestock and cropland to mitigate agricultural pollution in China. Resources, Conservation and Recycling 144, 65-73 (2019).

REVIEWER COMMENTS

Reviewer #1 (Remarks to the Author):

I thank the authors for addressing the questions I had very clearly. In particular, I appreciated the addition of GHG benefits as this adds a valuable component to the results. I am happy to support the acceptance of this manuscript for publication. I suggest three minor edits

- With respect to my previous comment 3) 'the land area released from the reduction of growing animal feed is much larger, the net land requirement for crop and livestock production under DIET scenario decline', I think it would be useful to add this short explanation to the main text to aid the reader. Perhaps this could be added to Table 3?

- Table 2, The mitigation potential column does not need a % symbol in each row (unless this is style related)

- Table 3 - capitalise consistently in the far right column.

Reviewer #2 (Remarks to the Author):

I still worry about the reliability of photochemical modeling results because most of the conclusions were based on them. I did not find any evaluations on secondary nitrate and secondary sulfate simulations, which depend strongly on ammonia emissions. Please add them in the revised manuscript.

Responses to Referees

Reviewer #1 (Remarks to the Author):

I thank the authors for addressing the questions I had very clearly. In particular, I appreciated the addition of GHG benefits as this adds a valuable component to the results. I am happy to support the acceptance of this manuscript for publication. I suggest three minor edits

We appreciate the reviewer for constructive comments and valuable suggestions, which have helped us improve the manuscript.

- 1) With respect to my previous comment 3) 'the land area released from the reduction of growing animal feed is much larger, the net land requirement for crop and livestock production under DIET scenario decline', I think it would be useful to add this short explanation to the main text to aid the reader. Perhaps this could be added to Table 3?

Good suggestion. We have added this short explanation in Table 3 (Line 563).

- 2) Table 2, The mitigation potential column does not need a % symbol in each row (unless this is style related)

Revised as suggested (Line 534).

- 3) Table 3 - capitalise consistently in the far right column.

Revised as suggested.

Reviewer #2 (Remarks to the Author):

I still worry about the reliability of photochemical modelling results because most of the conclusions were based on them. I did not find any evaluations on secondary nitrate and secondary sulfate simulations, which depend strongly on ammonia emissions. Please add them in the revised manuscript.

We acknowledge the reviewer's concern. The WRF-CMAQ model has been frequently used for the PM_{2.5} forecasts and policy support over the East Asian countries to deal with environmental issues associated with high levels of atmospheric PM_{2.5} concentrations (Choi et al., 2019; Xu et al., 2017; Hu et al., 2017; Wong et al., 2012). We have added discussion on the reliability of photochemical modelling results in the revised SI as follow: "The correlation coefficient between observed and simulated annual average PM_{2.5} concentration was 0.82 (n=302, p<0.05), with normalized mean biases (NMB) of -21.67 and normalized mean error (NME) of 29.49 (new Table S23). From seasonal perspective, the simulated proportions of sulfate, nitrate, and ammonium in spring (April) and autumn (October) agreed well with observed values (new Fig. S12), but they were slightly underestimated in winter (January) and slightly overestimated in Summer (July). The possible reasons could be bias in the representation of heterogeneous chemical reaction in the WRF-CMAQ model or bias in the emission inventory and meteorological variables (wind speed, temperature, precipitation) simulation, etc (Choi et al., 2019). Such biases in input parameters and the results are inevitable, but on the whole and annual scale, the WRF-CMAQ model is still able to

reproduce spatial secondary inorganic aerosols and disentangle the effects of NH₃ mitigation on the PM_{2.5} pollution. More research is needed to improve the model parameterization and performance. This is beyond the scope of our study, which focuses on the assessment of NH₃ mitigation potential and cost-effective mitigation strategies.” (SI Lines 478-491)

[REDACTED]

Table S23 Performance statistics of modelling data and monitoring data of PM_{2.5}

	n	r	NEB	NMB
January	302	0.75	-24.76	33.18
April	302	0.72	-33.44	37.98
July	302	0.73	-1.00	36.86
October	302	0.75	-20.61	30.53
Annual average	302	0.82	-21.97	29.49

Evaluation on secondary nitrate and secondary sulfate simulations has been added in the revised SI as follow: “Responses of different compositions of PM_{2.5} to NH₃ mitigation were investigated by WRF-CMAQ model simulation. Results showed that reducing NH₃ emissions could significantly decrease the annual average concentration of nitrate, ammonium and PM_{2.5} concentrations (new Fig. S13). Sulfate concentration is not sensitive to NH₃ mitigation (Fig. R1) because of its low vapor pressure and thermodynamic stable nature (Wang et al., 2016; Liu et al., 2019). In contrast, nitrate concentration is very sensitive to changes in NH₃ emissions (Fig R2) because of its high saturated vapor pressure and thermal stability (Xu et al., 2019). The availability of atmospheric NH₃ is one of the key factors determining the transformation of HNO₃ into NH₄NO₃ in China. This model result is also consistent with Liu et al. (2019), Xu et al. (2019) and our ongoing research simulated by WRF-Chem model (Fig

R3. (SI Lines 493-501)

Figure S13 | Responses of annual concentration of PM_{2.5} compositions to NH₃ mitigation in China simulated by WRF-CMAQ (modified from Xu et al. 2017)

[REDACTED]

[REDACTED]

Figure R3 | Monthly concentration of PM_{2.5} compositions to NH₃ mitigation in BTH (Beijing–Tianjin–Hebei) simulated by WRF-Chem model (ongoing research)

Reference:

- (1) Wong, D.C., et al. WRF-CMAQ two-way coupled system with aerosol feedback: software development and preliminary results. *Geosci. Model Dev.* 5, 299-312 (2012).
- (2) Xu, Y., et al. Sensitivity analysis of PM_{2.5} pollution to ammonia emission control in China. *China Environ. Sci.* 37, 2482-2491 (2017).
- (3) Xu, Y., et al. Impact of Meteorological Conditions on PM_{2.5} Pollution in China during Winter. *Atmosphere* 9, 429 (2018).

- (4) Choi, M., Lee, J., Woo, J., Kim, C. & Lee, S. Comparison of PM_{2.5} Chemical Components over East Asia Simulated by the WRF-Chem and WRF/CMAQ Models: On the Models's Prediction Inconsistency. *Atmosphere* 10, 618 (2019).
- (5) Wang, G., et al. Persistent sulfate formation from London Fog to Chinese haze. *Proc. Natl Acad. Sci. USA*, 201616540 (2016).
- (6) Liu, M., et al. Ammonia emission control in China would mitigate haze pollution and nitrogen deposition but worsen acid rain. *Proc. Natl Acad. Sci. USA*, 201814880 (2019).
- (7) Xu, Z., et al. High efficiency of livestock ammonia emission controls in alleviating particulate nitrate during a severe winter haze episode in northern China. *Atmos. Chem. Phys.* 19, 5605-5613 (2019).

REVIEWERS' COMMENTS:

Reviewer #2 (Remarks to the Author):

This manuscript could be accepted for publication.

Response to Reviewers' Comments:

Reviewer #2 (Remarks to the Author):

This manuscript could be accepted for publication.

We appreciate the reviewer for constructive comments and valuable suggestions, which have helped us improve the manuscript.